# Shortening the Lipid A Acyl Chains of *Bordetella pertussis* Enables Depletion of Lipopolysaccharide Endotoxic Activity

**DOI:** 10.3390/vaccines8040594

**Published:** 2020-10-09

**Authors:** Jesús Arenas, Elder Pupo, Coen Phielix, Dionne David, Afshin Zariri, Alla Zamyatina, Jan Tommassen, Peter van der Ley

**Affiliations:** 1Department of Molecular Microbiology and Institute of Biomembranes, Utrecht University, 3584 CH Utrecht, The Netherlands; J.P.M.Tommassen@uu.nl; 2Unit of Microbiology and Immunology, Faculty of Veterinary, University of Zaragoza, 500017 Zaragoza, Spain; 3Institute for Translational Vaccinology (Intravacc), 3721 MA Bilthoven, The Netherlands; elder.pupo.escalona@intravacc.nl (E.P.); coen.phielix@intravacc.nl (C.P.); Dionne.david@intravacc.nl (D.D.); afshin.zariri@intravacc.nl (A.Z.); peter.van.der.ley@intravacc.nl (P.v.d.L.); 4Department of Chemistry, University of Natural Resources and Life Sciences, 1190 Vienna, Austria; alla.zamyatina@boku.ac.at

**Keywords:** *Bordetella pertussis*, LPS, lipid A engineering, endotoxin, whole-cell vaccine, reactogenicity

## Abstract

Whooping cough, or pertussis, is an acute respiratory infectious disease caused by the Gram-negative bacterium *Bordetella pertussis.* Whole-cell vaccines, which were introduced in the fifties of the previous century and proved to be effective, showed considerable reactogenicity and were replaced by subunit vaccines around the turn of the century. However, there is a considerable increase in the number of cases in industrialized countries. A possible strategy to improve vaccine-induced protection is the development of new, non-toxic, whole-cell pertussis vaccines. The reactogenicity of whole-cell pertussis vaccines is, to a large extent, derived from the lipid A moiety of the lipopolysaccharides (LPS) of the bacteria. Here, we engineered *B. pertussis* strains with altered lipid A structures by expressing genes for the acyltransferases LpxA, LpxD, and LpxL from other bacteria resulting in altered acyl-chain length at various positions. Whole cells and extracted LPS from the strains with shorter acyl chains showed reduced or no activation of the human Toll-like receptor 4 in HEK-Blue reporter cells, whilst a longer acyl chain increased activation. Pyrogenicity studies in rabbits confirmed the in vitro assays. These findings pave the way for the development of a new generation of whole-cell pertussis vaccines with acceptable side effects.

## 1. Introduction

The Gram-negative bacterium *Bordetella pertussis* is an obligate human pathogen that causes pertussis, an acute respiratory tract disease also known as whooping cough. Several vaccine formulations have been developed against pertussis. A whole-cell pertussis vaccine that was introduced in the fifties of the previous century was effective but generated significant side effects. Therefore, it was replaced by subunit-based vaccines, which were shown to be safe and effective in inducing protective immunity [1]. However, particularly the industrialized countries, where these vaccines are in use, have witnessed a resurgence of pertussis in the past decades [2]. This could be related to the rapid waning of the immunity induced by the acellular vaccines, which evoke a T-helper (Th) 2 response in contrast to the long-lived Th1/Th17-skewed immunity elicited by whole-cell vaccines or by natural infection [3]. Moreover, in contrast to the whole-cell vaccines, the acellular vaccines do not prevent colonization of the upper respiratory tract and, thus, transmission from vaccinees to unprotected individuals [4,5]. Vaccine-driven evolution of circulating *B. pertussis* strains may also contribute to the resurgence of pertussis incidence [6,7]. Thus, there is a strong medical need for a new, safe, and effective vaccine formula. A strategy to reach this goal could be the introduction of new whole-cell vaccines with reduced toxicity. As toxicity is mainly determined by the lipid A moiety of lipopolysaccharides (LPS) [8], this approach requires lipid A engineering.

LPS is a major component of the outer membrane of Gram-negative bacteria. It consists of a lipid A moiety, a core oligosaccharide, and a long polysaccharide known as the O-antigen, which is lacking in some species including *B. pertussis* [9,10,11]. The lipid A moiety is recognized by the mammalian LPS receptor, the Toll-like receptor 4 (TLR4)-myeloid differentiation factor 2 (MD-2) complex, resulting in activation of a signaling cascade that ends in the production of pro-inflammatory cytokines and chemokines [12]. These mediators activate the immune defenses [13,14], but overstimulation causes a variety of disorders with often fatal consequences [15]. Thus, LPS can act as an adjuvant but also as a potent endotoxin.

The prototype lipid A of *Escherichia coli* consists of a glucosamine disaccharide that is phosphorylated at positions 1 and 4΄ and contains four 3-hydroxylated fatty acyl chains linked via an amide linkage to positions 2 and 2′ and via an ester bond to positions 3 and 3′ (Figure 1A). Two secondary acyl chains are attached at the hydroxyl groups of the fatty acids at positions 2′ and 3′ [10]. The biosynthetic pathway of lipid A requires nine well-conserved enzymes [10]. In the first step, a 3-hydroxyl (3OH) acyl chain is transferred from acyl carrier protein to the 3 positions of N-acetylglucosamine (GlcNAc) in the activated sugar UDP-GlcNAc by LpxA [16,17]. The resulting product is then de-acetylated by LpxC and subsequently acylated with a 3OH acyl chain at position 2 by LpxD. LpxH then removes a UMP molecule from a portion of the resulting molecules and one modified molecule is linked with an unmodified one by LpxB. The resulting product is phosphorylated at position 4′ by LpxK to create the tetra-acylated and bis-phosphorylated lipid IV_A_. Two 3-deoxy-D-*manno*-oct-2-ulosonic acid (KDO) residues are then added to position 6′ by WaaA after which the secondary acyl chains are added by the LpxL and LpxM acyltransferases.

Variations in the lipid A structure are found in different bacterial species. These variations affect the activation of the LPS receptor. Particularly, the number and length of the acyl chains as well as the number and modifications of the phosphate groups determine the strength of activation [10,18,19,20]. Variation in the acyl-chain length is determined by molecular rulers in the acyltransferases LpxA, LpxD, LpxL, and LpxM, which vary between these enzymes of different bacterial species [21,22]. Furthermore, after the conserved biosynthesis pathway, modifications can be introduced in the lipid A during or after its transport to the outer membrane by enzymes located in the inner or outer membrane. These modifications include acylation, de-acylation, de-phosphorylation, and modification of the phosphoryl groups, and the presence of the corresponding enzymes differs between bacterial species [23]. Thus, lipid A engineering is possible by exploiting the variation between different species [20,24,25].

Lipid A of *B. pertussis* differs from that of *E. coli* in that it is penta-acylated (Figure 1A); it misses a secondary acyl chain linked to the primary acyl chain at the position 3′. Furthermore, the remaining secondary acyl chain is a C14 instead of a C12 as found in *E. coli* and, whilst the primary acyl chain at position 3′ is 3OH-C14 as in *E. coli*, there is a 3OH-C10 at position 3 (Figure 1A) even though these acyl chains are added by the same LpxA enzyme. As we recently demonstrated, this asymmetry is caused by the low substrate selectivity of LpxA in combination with the substrate specificity of LpxH in *B. pertussis* [26]. In this study, we investigated the possibility to reduce the toxicity of *B. pertussis* LPS by exploiting the different chain-length specificities of the LpxA, LpxD, and LpxL enzymes from different bacterial species.

## 2. Material and Methods

### 2.1. Plasmids, Strains, and Growth Conditions

Table 1 lists all plasmids and strains used in this study. *B. pertussis* strains B213 [27] and B1917 [28] were cultured on Bordet-Gengou agar (Difco, Le Pont de Claix, France) supplemented with 15% defibrinated sheep blood (Biotrading, Mijdrecht, The Netherlands) for 48 h at 35 °C. To grow the bacteria in liquid cultures, bacteria were collected from solid medium and diluted in Verweij medium [29] to an optical density at 590 nm (OD_590_) of 0.05 and incubated in 125-mL square bottles with constant shaking at 175 rpm. For some assays, the bacteria were inactivated by incubation for 1 h at 60 °C, resuspended in phosphate-buffered saline (137 mM NaCl, 2.7 mM KCl, 4.3 mM Na_2_HPO_4_, 1.47 mM KH_2_PO_4_, pH 7.4) (PBS) and adjusted to an OD_590_ of 0.5. *E. coli* strains were grown in lysogeny broth (LB) or LB agar at 37 °C.

For all strains, media were supplemented with kanamycin (100 μg ml^−1^), gentamicin (10 μg mL^−1^), ampicillin (100 μg mL^−1^), nalidixic acid (50 μg mL^−1^), or streptomycin (300 μg mL^−1^) when required for selection or plasmid maintenance, and with 0.1 or 1 mM isopropyl-β-D-1-thiogalactopyranoside (IPTG) to induce gene expression in *E. coli* or *B. pertussis*, respectively.

### 2.2. Genetic Manipulations

PCRs were performed using High Fidelity Polymerase (Roche Diagnostics GmbH, Mannheim, Germany). PCR mixes consisted of 1 μL of genomic DNA as a template, 200 µM dNTPs (ThermoFisher Scientific, Vilnius, Lithuania), 0.25 µM of different primer combinations, 0.5 U DNA polymerase, and PCR buffer. All primers used are listed in Appendix A. The mixtures were incubated for 10 min at 95 °C for DNA denaturation, followed by 30 cycles of 1 min at 95 °C, 0.5 min at 58 °C and elongation at 72 °C for 1 min per kb of expected amplicon size. The reactions were terminated with an extended elongation step for 10 min at 72 °C. The resulting products were separated on 1% agarose gels by electrophoresis and visualized using ethidium bromide.

Genes encoding LPS-biosynthesis enzymes of different bacteria were amplified by PCR and cloned into broad host-range expression vector pMMB67EH [30] by exchanging them for the *pagL* gene in pMMB67EH-PagL_Pa_ [8] (Table 1). To this end, the PCR products and plasmid pMMB67EH-PagL_Pa_ were purified using the Clean-Up System and Plasmid Extraction kit (both from Promega, Madison, USA), respectively. Purified plasmid and PCR products were digested for 2 h at 37 °C with NdeI and HindIII (ThermoFisher Scientific) for which recognition sites are included in the PCR primers (Appendix A), and subsequently ligated together overnight at room temperature using T4 DNA ligase (ThermoFisher Scientific). The resulting products were used to transform *E. coli* strain DH5α, and clones containing the target gene inserted in the pMMB67EH backbone were elected by restriction enzyme digestions and DNA sequencing at Macrogen (Amsterdam, The Netherlands).

The *lpxL*_2_ gene is located in an operon of three genes in between the *lpxL_1_* and *dapF* genes. To inactivate *lpxL_2_* by allele exchange, we made use of the suicide vector pKAS32 [31] (Table 1). Initially, plasmid pKA32-EFGH-*lpxL*_2_::*gm* was used (Table 1). This plasmid contains a 750-bp sequence of the 3′ end of the *lpxL*_1_ gene and a 753-bp sequence immediately downstream of *lpxL*_2_ separated by a gentamicin-resistance cassette. Attempts to knock out the *lpxL_2_* gene with this construct failed. We considered the possibility that the rigorous deletion of the *lpxL*_2_ gene might affect the expression of the downstream-located *dapF* gene. Therefore, constructs with a more subtle deletion, affecting only the first two codons of *lpxL*_2_, were generated. The downstream sequence in pKA32-EFGH-*lpxL*_2_::*gm* was replaced by a PCR product obtained with primers listed in Appendix A and encompassing the *lpxL*_2_ gene but without the first six nucleotides and, in addition, 866 bp downstream of *lpxL_2_*. Then, a gentamicin-resistance cassette, also obtained by PCR, was re-introduced in the resulting plasmid, and plasmids pKO*lpxL*_2_::*gm* (a) and pKO*lpxL*_2_::*gm* (b), which contain the resistance cassette in two different orientations, were selected (Table 1).

Plasmid pRT669 was used to knock out the *B. pertussis lpxA* gene. *P. aeruginosa lpxA* and *lpxD* genes were used to replace the *B. pertussis* counterparts by homologous recombination as follows. Synthetic DNA fragments constituting the *fabZ*-*lpxA*-*lpxB* region (1599 nucleotides) and the *skp-lpxD-fabZ* region (1881 nucleotides) were designed, such that the *lpxA* and *lpxD* open reading frames were replaced by the *P. aeruginosa* sequence while the flanking regions were derived for *B. pertussis*. The synthetic DNAs were obtained from Genscript and cloned into the HindIII site of the suicide shuttle vector pSS1129 [32]. The resulting plasmids were called pC*lpxA*_Pa_ and pC*lpxD*_Pa_.

*E. coli* DH5α was transformed with ligation products or plasmids following standard protocols. Correct clones were elected by PCR, and plasmids were purified and sequenced at the Macrogen sequencing service (Amsterdam). Then, plasmids were transferred to *E. coli* strain SM10λpir [33] by transformation and subsequently to *B. pertussis* strain B213 by conjugation using ampicillin and nalidixic acid for selection and counterselection, respectively. To generate chromosomal mutations, various knockout plasmids (Table 1), which contain an *rpsL* gene conferring streptomycin sensitivity [31], were integrated into the chromosome via a single crossover by selecting for kanamycin- or gentamicin-resistant transconjugants; the resulting bacteria had lost streptomycin resistance. Subsequently, to select for plasmid loss by a second crossover, bacteria were grown in liquid medium and mutants were selected on plates with streptomycin and kanamycin or gentamicin. The presence of the plasmids in *B. pertussis* transconjugants and the proper generation of knockout mutants were verified by PCR.

### 2.3. RNA Extraction and Reverse Transcriptase (RT)-PCR

To obtain RNA, cells from exponentially growing cultures were collected by centrifugation for 10 min at 5000 rpm in an Eppendorf Centrifuge 5424 (Eppendorf, Nijmegen, The Netherlands), adjusted to an OD_550_ of 4, and resuspended in trizol (Invitrogen, Carlsbad, USA). Then, 200 μL of chloroform were added per ml of trizol, followed by centrifugation at 5000 rpm for 30 min. The resulting upper layer was mixed with an equal volume of ice-cold 75% ethanol. Next, RNA was isolated using the Nucleospin RNA II kit (Macherey-Nagel GmbH, Düren, Germany) according to the manufacturer’s instructions. The resulting solution was treated with Turbo DNA free (Ambion, Berlin, Germany) for 1 h at 37 °C to remove genomic DNA followed by DNase inactivation according to recommendations of the manufacturer to generate pure RNA. This was used immediately to generate cDNA using the Transcriptor High Fidelity cDNA Synthesis Kit (Roche, Amsterdam, The Netherlands). RNA, cDNA and genomic DNA were used as templates in PCRs with primers listed in Appendix A to determine the generation of specific transcripts following previously described protocols [34].

### 2.4. Electrophoresis and Western Blotting

Whole-cell lysates from bacteria adjusted to an OD_600_ of 5.0 were mixed 1:1 with double-strength sample buffer, and heated for 10 min at 100 °C. For the detection of LPS, whole-cell lysates were boiled in sample buffer as above and then treated with proteinase K for 1 h at 37 °C. Proteins and LPS were separated by SDS-PAGE on gels containing 13% and 16% acrylamide, respectively, after which they were stained with Coomassie brilliant blue G250 or silver, respectively. For Western blotting, proteins separated on gels were transferred to nitrocellulose membranes as described [35] and anti-6xHis tag antibodies (Invitrogen) were used for protein detection.

### 2.5. LPS Purification and Analysis

For LPS isolation, bacteria were grown to the exponential phase, i.e., for 12 h for wild type and chromosomal mutant derivatives and for 24 h for strains B213 Δ*lpxA*-pLpxA_Pa_, B213-pLpxD_Pa_, and B213-pLpxL_Nm_. LPS were extracted with hot phenol-water [36] and purified further by solid-phase extraction on C8 reversed-phase cartridges as described [26]. Negative-ion ESI Fourier transform (FT) mass spectrometry (MS) of purified LPS was performed on an LTQ Orbitrap XL instrument (ThermoFisher Scientific) following the described methods [26]. LPS samples were dissolved in a mixture of 2-propanol, water, and triethylamine (50:50:0.001, *v*/*v*/*v*) (pH 8.5) and infused into the mass spectrometer by nano-electrospray ionization (ESI)-mass spectrometry (MS) using gold-coated pulled glass capillaries. The spray voltage was set to from −1.3 to −1.85 kV and the temperature of the heated capillary to 250 °C. Then, nano-ESI-FT-MS was performed with in-source collision-induced dissociation [26]. This fragmentation technique produced intense fragment ions corresponding to intact lipid A domain, which originated from the rupture of the labile linkage between the nonreducing lipid A glucosamine and 3-deoxy-D-*manno*-oct-2-ulosonic acid. Lipid A compositions proposed are based on the chemical structure of the LPS from *B. pertussis* reported previously [37]. Mass-to-charge ratios given refer to mono-isotopic molecular masses.

### 2.6. Eukaryotic Cell Lines and TLR4 Signalling

Human NF-κB/SEAP reporter HEK293-Blue cells stably transfected with human TLR4 (hTLR4), mouse TLR4 (mTLR4) in combination with the corresponding MD-2 and CD14 or human TLR2 (hTLR2) were purchased from InvivoGen. These cell lines contain an NF-κB-inducible secreted embryonic alkaline phosphatase (SEAP) reporter gene. The cells were grown in DMEM supplemented with 10% (*v*/*v*) heat-inactivated fetal bovine serum (FBS), 100 U/ml penicillin, 100 µg mL^−1^ streptomycin, 300 ng mL^−1^ L-glutamine, 100 µg mL^−1^ normocin, and 1× HEK Blue selection at 37 °C in a 5% saturated CO_2_ atmosphere. For TLR4 signaling, HEK-Blue cells (2.5 × 10^4^) were incubated with serial dilutions of purified LPS or heat-inactivated bacterial cell preparations adjusted to an OD_590_ of 0.1 in a 96-well plate as before [26]. As a positive control, ultrapure LPS from *E. coli* K-12 (InvivoGen, Toulouse, France) was used. After 16–24 h incubation at 37 °C, supernatants were collected and incubated for 2 h with the SEAP substrate Quanti-Blue (InvivoGen), and the OD_630_ was measured using an absorbance microplate reader.

### 2.7. Animal Experiments

The animal experiments of this study were performed at Triskelion B.V., Zeist, The Netherlands. The welfare of the animals was maintained in accordance with the general principles governing the use of animals in experiments of the European Community and Dutch legislation. This included licensing of the project by the Central Committee on Animal Experimentation (project license 2016602) and approval of the study by the Triskelion Animal Welfare Body (AWB number TRIS-333). Five male New Zealand rabbits per group were housed and manipulated according to standard operating procedures of Triskelion facilities. The body temperature of each animal was recorded three times a day for five days before the first injection, just before the administration, and about 0.5, 1, 2, 4, 6, 24, and 48 h after the injection. The body temperature was measured using an external scanner from subcutaneously implanted transponders (Plexx, IPTT-300 transponder, Elst, The Netherlands). The rabbits were inoculated intramuscularly with a saline solution or 10 µg of pure LPS in saline solution. All rabbits were considered in the final data set.

### 2.8. Statistical Analysis

Body temperature data before and after the treatment were statistically analyzed with one-way analysis of variance (ANOVA) and Dunnett tests. For pre-treatment analysis, a generalized ANOVA test with automatic data transformation was used. This test is an automatic decision tree consisting of: (i) data pre-processing test, (ii) a group test assessing whether or not group test means were all equal, and (iii) post-hoc analysis. If the group test showed significant (*p* < 0.05) non-homogeneity of group means, pairwise comparisons with the control group were conducted by Dunnett’s multiple comparison test (parametric after ANOVA, non-parametric after Kruskal–Wallis; significance levels 0.01 and 0.05). The post-treatment analyses were performed as the pre-treatment with some modifications. For the pre-processing test, the suitability of the covariate was checked (criteria: sufficient cases, at least 2; variability of covariate non-zero; covariate effects sufficiently parallel over the groups; significance level parallelism test 0.01). Next, normality of data distribution (Shapiro–Wilks test; significance level 0.05) and homogeneity of variances (Levene test; significance level 0.05) were checked. If the group test showed significant (*p* < 0.05) non-homogeneity of group means, pairwise comparisons with the control group were conducted by Dunnett’s multiple comparison test (parametric after ANOVA, non-parametric after Kruskal–Wallis; significance levels 0.01 and 0.05). Besides, areas under the curve between particular intervals were calculated for each rabbit considering the mean of all rabbits of each group as the baseline. Then, the mean and standard deviation of the area under the curve was calculated for each group and utilized for comparisons between two groups using an unpaired *t*-test. Statistical analysis and comparisons were calculated using the GraphPad software version 6.01.

## 3. Results

### 3.1. Production of Heterologous Lpx Enzymes in B. pertussis

To modify the length of the primary acyl chains at positions 2, 2′, and 3, and of the only secondary acyl chain in *B. pertussis* lipid A, we made use of LpxD, LpxA, and LpxL acyltransferases from other bacteria. *B. pertussis* lipid A contains 3OH-C10 and 3OH-C14 chains at positions 3 and 3′, respectively (Figure 1A). Substitution of LpxA by the corresponding enzyme from *Pseudomonas aeruginosa* (LpxA_Pa_) would be expected to result in 3OH-C10 chains at both positions. Substitution of LpxD of *B. pertussis* by that of *P. aeruginosa* (LpxD_Pa_) was expected to lead to the substitution of the 3OH-C14 chains at positions 2 and 2′ by 3OH-C12 chains. Finally, the substitution of LpxL by the corresponding enzymes from *Porphyromonas gingivalis* (LpxL_Pg_) or *Neisseria meningitidis* (LpxL_Nm_) would be expected to result in the replacement of the secondary C14 chain by C16 or C12 chains, respectively. The expected structures are illustrated in Figure 1B.

The genes for the heterologous enzymes were cloned into the broad host-range expression vector pMMB67EH under the control of the *tac* promoter. Gene expression was first evaluated in the cloning host *E. coli* BL21(DE3) by RT-PCR. These assays confirmed the presence of transcripts of the genes of interest when the bacteria were grown with IPTG, whilst these transcripts were much less abundant or undetectable when the bacteria were grown in the absence of IPTG (Appendix A and results not shown). Synthesis of the proteins was also detected by SDS-PAGE; LpxA_Pa_ was produced in higher amounts than the LpxL_Nm_ and LpxL_Pg_ proteins (Appendix A). LpxD_Pa_ protein was not detected on a stained SDS-PAGE gel. To facilitate its detection, a 6xHis-tag was engineered at the C terminus of the protein. Western blotting assays showed a band of the expected size after expressing *lpxD*_Pa_ in *E. coli* BL21(DE3) by the addition of IPTG (Appendix A). The plasmids were then transferred to *B. pertussis* strain B213. The synthesis of LpxL_Nm_ and LpxD_Pa_ in B213 impaired growth, whilst the strains producing LpxA_Pa_ or LpxL_Pg_ grew like the wild type (Appendix A).

### 3.2. Analysis of Recombinant Lipid A Structures

The lipid A structures were analyzed by nano-ESI-MS analysis using purified LPS extracted from exponentially growing bacteria in the presence of IPTG. For the wild-type strain, a major peak was observed at *m/z* 1557.97, which corresponds with the expected *bis*-phosphorylated penta-acylated lipid A (Figure 2A). In the strain expressing LpxA_Pa_, the spectrum revealed, besides the ion at *m/z* 1557.97, two additional prominent ions at *m/z* 1501.91 and 1529.94 (Figure 2B). The ion at *m/z* 1501.91 corresponds with the expected substitution of the primary 3OH-C14 acyl chain at position 3′ by 3OH-C10, whilst the *m/z* 1529.94 ion indicates its substitution by a hydroxylated fatty acid with an intermediary C12 chain length. The relative abundance of the two new species was only 75 and 48%, respectively, relative to the wild-type structure at *m/z* 1557.97, which could be due to the expression of the endogenous *lpxA* on the chromosome. Therefore, we decided to knock out the chromosomal *lpxA* copy. This inactivation caused a growth defect (Appendix A), perhaps due to a polar effect on the expression of the downstream located *lpxB* gene. MS analysis of the lipid A of the resulting strain showed the complete loss of the *m/z* 1557.97 ion and a drastic decrease in the abundance of the *m/z* 1529.94 ion leaving a major peak of *m/z* 1501.91 corresponding to the expected substitution (Figure 2C).

MS analysis of lipid A from B213-pLpxL_Nm_ showed a drastic reduction of the wild-type *m/z* 1557.97 ion, whilst the major peak at *m/z* 1529.94 corresponds with the expected substitution of the secondary C14 acyl chain by C12 (Figure 2D). Attempts to delete the chromosomal *lpxL* failed*. B. pertussis* contains two adjacent *lpxL* homologs on the chromosome, but only one of them, called *lpxL_2_*, is active under laboratory growth conditions [38]. Different constructs were used to delete the *lpxL_2_* gene partially or completely; however, despite considerable efforts, we could not obtain the desired knockout. This is not due to a polar effect of *lpxL_2_* disruption on the expression of the downstream gene, *dapF*, which encodes an essential enzyme involved in the synthesis of *meso*-diaminopimelate (DAP), a precursor of the cell wall, and lysine, since also attempts to inactivate the *lpxL*_2_ gene in the presence of *meso*-DAP were unsuccessful. Thus, whilst the partial substitution of wild-type LPS in B213 expressing *lpxL*_Nm_ already causes a considerable growth defect (Appendix A), the complete substitution of this LPS by the altered form is possibly lethal.

MS analysis of lipid A from B213-pLpxL_Pg_ also revealed a drastic reduction in the abundance of the *m/z* 1557.98 ion and a new peak was detected in this case at *m/z* 1586.01, which corresponds with the expected substitution of the C14 by a C16 chain (Figure 2E). Peculiarly, although the vast majority of lipid A was substituted in this strain without causing any appreciable growth defect (Appendix A), inactivation of the chromosomal *lpxL_2_* gene appeared also not possible in this strain.

Analysis of the lipid A structure from B213-pLpxD_Pa_ revealed, besides the ion at *m/z* 1557.97 corresponding with the wild-type structure, two abundant ions at *m/z* 1529.94 and *m/z* 1501.91 (Figure 2F) corresponding with the reduction of the length of one or two acyl chains, respectively, from 3OH-C14 to 3OH-C12. These results show that indeed the production of LpxD_Pa_ modified the structure of lipid A as predicted, although not completely. In summary, the heterologous production of LpxA_Pa_, LpxL_Nm_, LpxL_Pg,_ and LpxD_Pa_ in *B. pertussis* resulted in the expected LPS alterations.

### 3.3. Differential Activation of TLR4 by the LPS Variants

We next investigated whether the altered structures of the LPS affect TLR4 signaling. To this end, purified LPS preparations were added to HEK293-Blue cells expressing hTLR4. After exposure, the activation of the receptor was evaluated by measuring the expression of a reporter gene encoding SEAP. Interestingly, LPS from B213-pLpxA_Pa_ stimulated hTLR4 much less than did LPS from the wild-type strain (Figure 3A). The residual activation observed was due to the expression of the chromosomal *lpxA* gene, as it was eliminated after the inactivation of this gene (Figure 3A). Thus, the length of the primary acyl chain at position 3′ is relevant for the activation of hTLR4 by *B. pertussis* LPS. Furthermore, we tested whether modification of the acyl chains at positions 2 and 2′ would have a similar outcome. HEK-Blue cells expressing hTLR4 showed no induction of SEAP activity after incubation with purified LPS from B213-pLpxD_Pa_ (Figure 3A). Thus, also reduction of the length of the acyl chains at positions 2 and 2′ of *B. pertussis* lipid A has a drastic impact on the activation of hTLR4. Finally, LPS from B213-pLpxL_Nm_ and B213-pLpxL_Pg_ reduced and increased hTLR4 activation, respectively (Figure 3A). Hence, stimulation of hTLR4 also correlates with the length of the secondary acyl chain in the order C16 > C14 > C12.

For comparison, we also measured the activity of some of the LPS preparations in stimulating the mTLR4. Stimulation of HEK293-Blue cells expressing mTLR4 with LPS preparations from wild-type strain B213 resulted in a stronger response than observed in the cells expressing hTLR4 (compare panels A and B in Figure 3). LPS preparations from B213 cells expressing the heterologous enzymes, including those expressing LpxL_Pg_ resulting in a secondary C16 chain, appeared slightly less effective in stimulating these cells (Figure 3B). Thus, activation of the human and mouse TLR4 is differently affected by the LPS modifications. However, when the chromosomal *lpxA* gene was inactivated in B213 expressing LpxA_Pa_, the resulting LPS was severely affected in its capacity to activate mTLR4 (Figure 3B). Apparently, the length of the primary acyl chain at position 3′ is critical in stimulating both hTLR4 and mTLR4.

Previously, we reported that the decreased toxicity of *B. pertussis* LPS that had lost the primary acyl chain at the 3 position due to the expression of a *pagL* gene encoding a lipid A de-acylase was nullified in whole-cell preparations by its increased release from the membranes [8]. As our goal is to generate a new cellular vaccine, we wished to determine also the biological activity of whole-cell preparations from cells producing the LPS variants with altered acyl-chain length. Synthesis of heterologous LPS biosynthetic enzymes in strain B213 affected the stimulation of HEK293-Blue cells expressing hTLR4 similarly in whole-cell and purified LPS preparations (compare panels A and C in Figure 3). Stimulation of HEK293-Blue cells expressing mTLR4 by whole-cell preparations was barely affected by the expression of the heterologous enzymes in B213 (Figure 3D). However, whole-cell preparations of the Δ*lpxA* mutant of B213 producing LpxA_Pa_ failed also to activate these cells (Figure 3D).

### 3.4. Construction of Stable LPS Mutants in a Vaccine Strain

To generate stable strains with lipid A modifications, the *lpxA* and *lpxD* genes of *B. pertussis* were replaced on the chromosome by the corresponding genes from *P. aeruginosa.* The replacement of both genes was performed in strain B1917, a worldwide predominant strain associated with the resurgence of pertussis since the 1990s [28]. MS spectra of lipid A of B1917 indicated a similar structure as that of B213. The correct replacement of both genes was confirmed in PCR assays. The mutant strains showed growth defects similar to those of strain B213 ∆*lpxA*-pLpxA_Pa_ and B213-LpxD_Pa_ (data not shown). In both cases, the gene substitution resulted in the synthesis of lipid A species with the expected shortened acyl chains as evidenced by MS (Figure 2G,H). Both mutants were grown for up to 10 passages in a liquid medium, and subsequent MS analysis revealed no alteration of the lipid A structure (data not shown), thus evidencing the stable production of modified lipid A. Interestingly, the growth rate of the B1917 *lpxA*_Pa_ mutant was restored to wild-type levels after the consecutive passages, and the resulting strain is, thus, more suitable for vaccine production. In contrast, the growth defect of mutant B1917 *lpxD*_Pa_ persisted after successive passages. The bioactivity of LPS preparations from the mutants was determined in HEK293-Blue cells expressing mTLR4 or hTLR4. In accordance with the results obtained with the B213 derivatives (Figure 3A,B), LPS preparations from B1917 *lpxA*_Pa_ did barely activate hTLR4 or mTLR4, whilst LPS from B1917 *lpxD*_Pa_ showed drastically lower activity than wild-type LPS (Appendix A). As a control, we also stimulated the reporter cell lines with hexa-acylated *E. coli* LPS and found that, as expected, it is a more potent hTLR4 agonist than the penta-acylated *B. pertussis* LPS, whilst mTLR4 was stimulated similarly by both LPS preparations (Appendix A).

### 3.5. LPS-induced Pyrogenicity Response in Rabbits

We next wanted to determine the endotoxicity of the mutant LPS species *in vivo*. The purified LPS used in the pyrogenicity experiment was first checked for activation of HEK293-Blue cells expressing hTLR2, which confirmed the absence of contaminating lipoproteins (data not shown). We opted for the rabbit pyrogenicity test, one of the methods accepted in most pharmacopeias to control for the presence of pyrogens. Most of the amino-acid differences in TLR4 among species are located in a hypervariable region located between residues 285 and 366, which is involved in ligand recognition. The amino-acid similarly shared with hTLR4 in this region is greater in rabbit TLR4 (57%) than in mTLR4 (48%) [39]. Thus, a rabbit model could be more relevant to predict pyrogenicity humans than a mouse model. Four groups of five New Zealand White rabbits were used in this experiment. The body temperature of each animal was monitored three times a day for 5 days before treatment. Animals showed a body temperature between 37.0–38.6 °C. No statistically significant differences were found between groups. Animals were then injected intramuscularly with physiological saline solution (control group) or purified LPS from wild-type B1917, B1917 *lpxD*_Pa_ or B1917 *lpxA*_Pa_. The body temperature was monitored before injection (time 0) and regularly during 6 h post-injection, and again at 24 and 48 h post-injection. The control group did not show significant alterations in the body temperature during the first 6 h and thereafter (Figure 4A). However, the body temperature of the group injected with wild-type LPS was increased 2 and 4 h post-injection (Figure 4A), and at 4 h post-injection, the difference was statistically significant compared to the control group (Figure 4B). At 6 h post-injection and thereafter, the body temperature had decreased again to similar levels as in the control group (Figure 4A). In the groups that received the mutant LPS species, no significant differences were found at any time compared to the control group, although the LPS of mutant B1917 *lpxD*_Pa_ slightly increased the temperature at 4 h post-injection compared to control (Figure 4A,B). A comparison of the areas under the curve between 0 h and 6 h confirmed statistically significant differences in response between the *lpxA* mutant LPS and the wild-type LPS (Figure 4C). In conclusion, in contrast to the wild-type LPS, both mutant LPS forms elicited no signs or minimal signs of pyrogenicity in rabbits, which is in agreement with the in vitro TLR4 stimulation assays.

## 4. Discussion

The reactogenicity of whole-cell pertussis vaccines has led to their replacement by subunit vaccines. However, these subunit vaccines suffer from several shortcomings, i.e., (i) although they elicit protective immunity, they do not prevent the colonization of the entire respiratory tract and transmission to unprotected individuals, (ii) the immunity elicited is rapidly waning, and (iii) because they consist of a limited number of antigens, vaccine-induced escape mutants might be selected. The whole-cell vaccines did not have these limitations but they were reactogenic. The development of less reactogenic whole-cell vaccines could offer a solution. LPS is, to a considerable extent, responsible for the toxicity of the cellular pertussis vaccines [8]. Previously, it has been demonstrated that the length of the acyl chains affects the toxicity of lipid A of *N. meningitidis* [19] and of synthetic lipid A analogs [40]. In the case of *N. meningitidis* lipid A, both increasing and reducing the length of the acyl chains at the 3 and 3′ positions resulted in reduced TLR4-stimulating activity [19]. Stöver et al. systematically altered the length of the secondary acyl chains in synthetic hexa-acylated lipid A analogs with three secondary acyl chains and reported that molecules with three secondary C10 chains had optimal activity, whereas both longer and shorter secondary chains reduced or eliminated the activity [40]. In the present study, we investigated whether modification of acyl-chain length in *B. pertussis* lipid A could also reduce LPS endotoxicity. We found that shortening of the length of all four acyl chains tested diminished the capacity of the LPS to activate hTLR4. Accordingly, an engineered increase in the length of one of the acyl chains enhanced the capacity to stimulate hTLR4. Thus, the length of acyl chains is relevant for the activity of *B. pertussis* LPS and, therefore, its manipulation can be a useful tool in the development of a new generation of cellular pertussis vaccines with impaired endotoxicity.

For the acyl chain at position 3′, our data can explain previous results by others. LPS of *B. pertussis* strain 18–323 and of two clinical isolates were reported to be far less effective in activating hTLR4 than that of reference strain Tohama I [41,42]. Lipid A of these strains deviates in two aspects from that of strain Tohama I, i.e., (i) it contains 3OH-C10 and 3OH-C12 acyl chains instead of 3OH-C14 at position 3′ due to an amino-acid substitution in the LpxA protein, and (ii) it lacks a non-stoichiometric modification of the phosphates with glucosamine that is observed in Tohama I depending on the culture conditions [41,42]. Both aspects were reported to contribute to the reduced capacity of this LPS to stimulate HEK-Blue cells expressing hTLR4 [43]. Here, we demonstrated that an acyl-chain alteration at the 3′ position influences the lipid A activity. Shortening of the acyl chains at the other positions also drastically reduced hTRL4 responses even in the presence of considerable amounts of wild-type LPS (Figure 2), suggesting that these novel LPS variants work as TLR4 antagonists.

Why could subtle variations in acyl-chain length exert such a dramatic effect on the ability of *B. pertussis* lipid A to activate TLR4? Canonical TLR4 agonists, such as the hexa-acylated *E. coli* lipid A, are captured by the hydrophobic binding pocket of MD-2 in a specific orientation with the proximal GlcN ring facing the dimerization interface (agonist pose). Five acyl chains are accommodated in the binding pocket of MD-2, whereas the sixth chain, the 2N-acyl chain, lies outside the pocket and drives TLR4 complex dimerization (Appendix A) [44,45]. In contrast, under-acylated lipid A variants, such as tetra-acylated lipid IVa or a synthetic analog of anti-endotoxic *R. sphaeroides* lipid A, Eritoran, are bound in an opposite orientation (rotation by 180°, antagonist pose) with four or five lipid chains, respectively, fully inserted in the binding pocket of MD-2 (Appendix A) [46,47]. Since neither acyl chain is exposed on the surface of MD-2, these lipid A variants cannot stimulate receptor complex dimerization and cannot initiate pro-inflammatory signaling [48]. It has been shown that the exposure of the 2*N-*acyl chain on the surface of MD-2 is facilitated by spatial rearrangement of the diglucosamine backbone of lipid A-induced by protein binding (Appendix A) and depends on the orientation of the MD-2-bound lipid A [49,50]. Presentation of the 2*N*-acyl chain on the surface of the protein is only possible if lipid A is bound in an “agonist orientation”—the one found for *E. coli* lipid A in the co-crystal structures with PDB code 3FXI and 3VQ1. The external location of the 2*N-*acyl chain is further stabilized by the LPS-induced rearrangement of the Phe126 loop of MD-2. Phe126 is exposed to solvent in the ligand-free state and with bound antagonist lipid A (Appendix A) which prevents receptor homodimerization [51], whereas it is directed inward upon binding of an agonist ligand [52,53]. The latter arrangement advocates receptor complex dimerization and the induction of signaling (Appendix A) [54].

The acylation pattern of penta-acyl *B. pertussis* lipid A is notorious for its reduced ability to activate TLR4 complex [20,55,56]. However, it is also known that negatively charged inner-core sugars present in *Ra*-LPS enhance the affinity for TLR4/MD-2 and contribute to establishing the dimerization interface with the second TLR4* complex by ionic interactions [57,58]. MD-2 discriminates both the acylation pattern and the length of acyl chains in lipid A (as well as their total hydrophobic volume), binding in two different orientations (+/−180°) of *B. pertussis* lipid A in the binding pocket of MD-2 is possible. Binding in the orientation with the proximal GlcN ring facing the dimerization interface would result in the TLR4 activation, whereas an opposite pose would provide antagonistic activity.

If penta-acyl B213 *Ra*-LPS binds via its lipid A region to the hTLR4/MD-2 complex in the agonist orientation, only four acyl chains can be inserted into the binding pocket of hMD-2 as compared to five chains in *E. coli* lipid A (Figure 5A,B). The hydrophobic volume provided by one C10 and three C14 chains is smaller compared to that of *E. coli* lipid A so that the lipid chains are loosely packed in the binding pocket which results in a lower affinity and less efficient binding as compared to *E. coli Ra*-LPS. Still, the exposure of the 2*N*-acyl chain on the surface of MD-2 and the correct positioning of the phosphate group at position 1 delivers sufficient hydrophobic and ionic attraction, respectively, for the interaction with the second TLR4 * complex resulting in receptor complex dimerization and induction of pro-inflammatory signaling. The increased length of the secondary C16 chain at position 2′ in the LPS of strain B213-pLpxL_Pg_ contributes to a more accurate fitting of acyl chains in the binding pocket of MD-2 in the agonist orientation (Figure 5C) which enhances affinity and results in higher TLR4-stimulating activity. A decreased length of acyl chains, such as in the mutants expressing *lpxA*_Pa_ or *lpxD*_Pa_, reduces the total hydrophobic volume of lipid chains, thereby diminishing the binding affinity to MD-2 in the agonist orientation. Such mutant LPS variants would likely bind in an opposite orientation (rotation of the glucosamine backbone of lipid A by 180°) with all acyl chains accommodated within the binding cavity of MD-2. Binding in the antagonist orientation (Figure 5D,E) prevents exposure of one acyl chain on the surface of MD-2 and thus impedes the dimerization and inhibits the pro-inflammatory signaling.

Interestingly, most modifications barely affected mTLR4 activation unless when the residual synthesis of wild-type LPS was prevented by inactivation of the chromosomal gene copies, as illustrated for *lpxA* and *lpxD*. Thus, whilst some novel LPS species appear to be antagonistic for hTLR4 activation, this does not seem to be the case for mTLR4 activation. Previous studies reported species-dependent differences regarding TLR4 activation [59]. Examples are lipid IVa [54,60] and the penta-acylated meningococcal LpxL1-variant LPS [61], which can act as antagonists for hTLR4, but as agonists for mTLR4. Species specificity is caused in part by the dissimilarities in the structure of h- and mMD-2. The binding pocket of mMD-2 is narrower and deeper than that of hMD-2, which is predictive for higher affinity binding of underacylated lipid A variants [54]. Besides, mMD-2 possesses less positively charged residues at the rim of the binding groove, which means that ionic interactions of lipid A phosphates are less important for binding to mMD-2. These features explain the higher activating potency of both wild types on mTLR4 compared to hTLR4. Importantly, these differences limit the extrapolation of data from experimental animals to humans in vaccine trials [61]. However, the novel LPS species resulting from the production of LpxA_Pa_ failed to activate both mTLR4 and hTLR4 in vitro and did not cause toxicity in rabbits as revealed in pyrogenicity assays. The growth properties and the apparent stability of expression of modified lipid A make these strains suitable for the production of whole-cell vaccines but also other LPS-containing vaccines, such as those based on outer membrane vesicles [62]. Further vaccine improvement is possible by combining the lipid A modifications with other mutations affecting, for example, expression of specific antigens or improving vesicle formation, and work along these lines is being pursued by us. Apart from the relevance of these results for vaccine development, they demonstrate that alteration of the 3′ acyl chain is a key feature for the activation of the LPS receptor of different mammalian species. A similar reduction of TLR activation in different species makes the LpxA_Pa_ mutant particularly applicable in vaccine development, as the results from animal experiments will be more predictive for the human situation.

## 5. Conclusions

Our approaches to reduce the endotoxicity of whole-cell *B. pertussis* vaccines by lipid A engineering were effective and resulted in at least two promising pertussis LPS mutants. After their modification at the chromosomal level, the strains expressed one single lipid A species and they appeared to be stable. A slight modification of the length of any of the fatty acyl chains of *B. pertussis* LPS drastically affects TLR4 signaling. To the best of our knowledge, alteration of lipid A acyl-chain length by manipulation of LpxD and LpxL with consequences for TLR4 signaling has not been reported before. Thus, we believe that our findings open new avenues to the generation of new cellular vaccines for *B. pertussis* and probably also for other Gram-negative bacterial pathogens. In addition, this approach can be used to modify and fine-tune the endotoxic activity of other LPS-containing vaccines such as those based on outer membrane vesicles.

## 6. Patents

Part of this work is included in a European patent application (reference number 17160604.9), filed by Intravacc, with JA, EP, PvdL, and JT as inventors.

## Figures and Tables

**Figure 1 vaccines-08-00594-f001:**
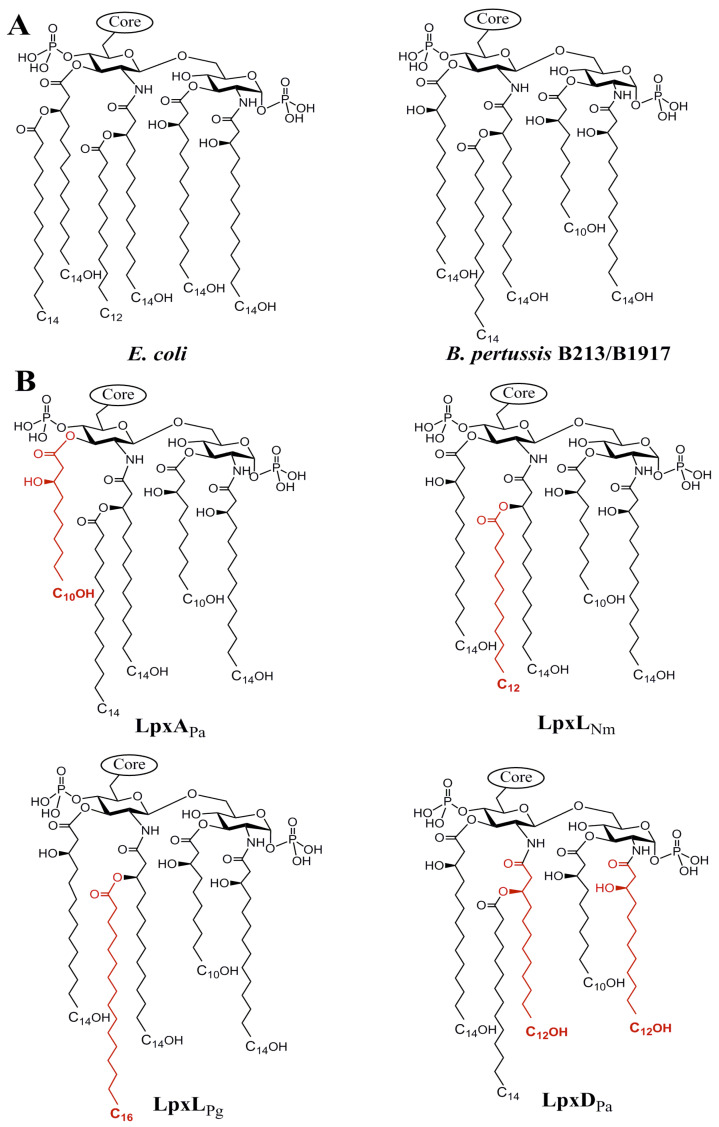
Lipid A structures of *Escherichia coli* and *Bordetella pertussis* (**A**) and those expected after replacing the acyltransferases in *B. pertussis* (**B**). The altered acyl chains that are introduced after replacing the acyltransferases of the wild type by the heterologous enzymes are shown in red. The heterologous enzymes expressed are indicated below the structures. Pa, *Pseudomonas aeruginosa*; Nm, *Neisseria meningitidis*; Pg, *Porphyromonas gingivalis*. 3OH-C14, 3OH-C12, and 3OH-C10 acyl chains are indicated as C14OH, C12OH, and C10OH, respectively.

**Figure 2 vaccines-08-00594-f002:**
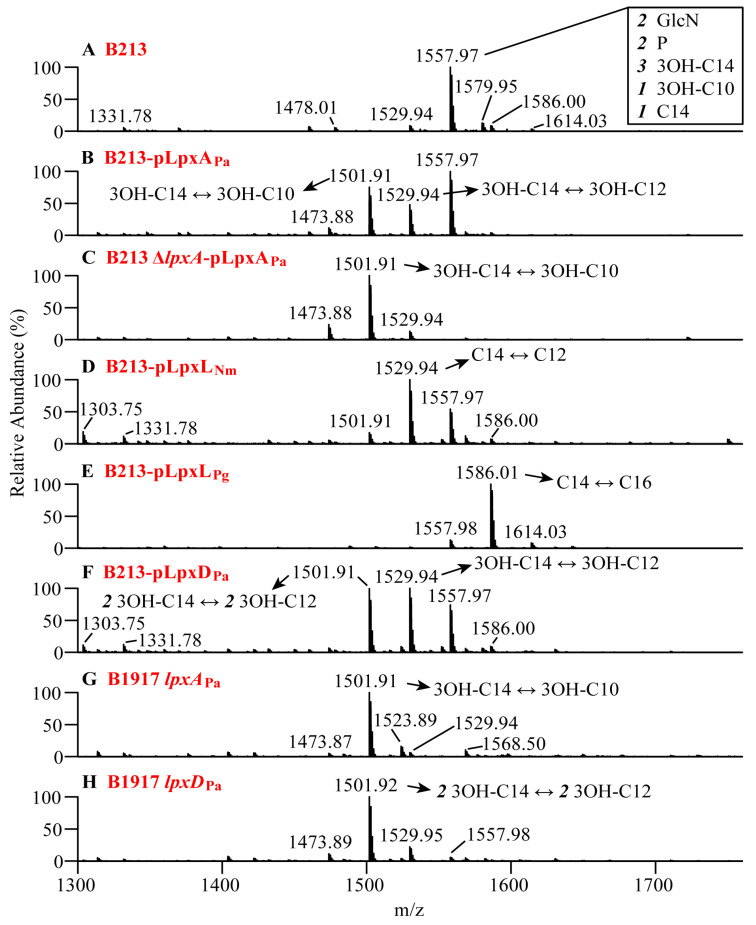
Structural analysis of lipid A by ESI-MS. Negative-ion lipid A mass spectra were obtained by in-source collision-induced dissociation nano-ESI-MS of intact LPS isolated from cells of (**A**) B213, (**B**) B213 expressing *lpxA*_Pa_ (B213-pLpxA_Pa_), (**C**) Δ*lpxA* mutant of B213 expressing *lpxA*_Pa_ (B213 Δ*lpxA*-pLpxA_Pa_), (**D**) B213 expressing *lpxL*_Nm_ (B213-pLpxL_Nm_), (**E**) B213 expressing *lpxL*_Pg_ (B213-pLpxL_Pg_), (**F**) B213 expressing *lpxD*_Pa_ (B213-pLpxD_Pa_), (**G**) B1917 with the chromosomal *lpxA* replaced by *lpxA*_Pa_ (B1917 *lpxA*_Pa_), and (**H**) B1917 with the chromosomal *lpxD* replaced by *lpxD*_Pa_ (B1917 *lpxD*_Pa_). A major singly-deprotonated ion at *m/z* 1557.97 was interpreted as the typical *B*. *pertussis* lipid A structure: a diglucosamine (2 GlcN), penta-acylated (three 3OH-C14, one 3OH-C10, and one C14) with two phosphates residues (2 P) as illustrated in Figure 1A. Additional singly-deprotonated lipid A ions were detected in different derivatives and their interpretations are also indicated. Only the *m/z* range covering lipid A ions is shown.

**Figure 3 vaccines-08-00594-f003:**
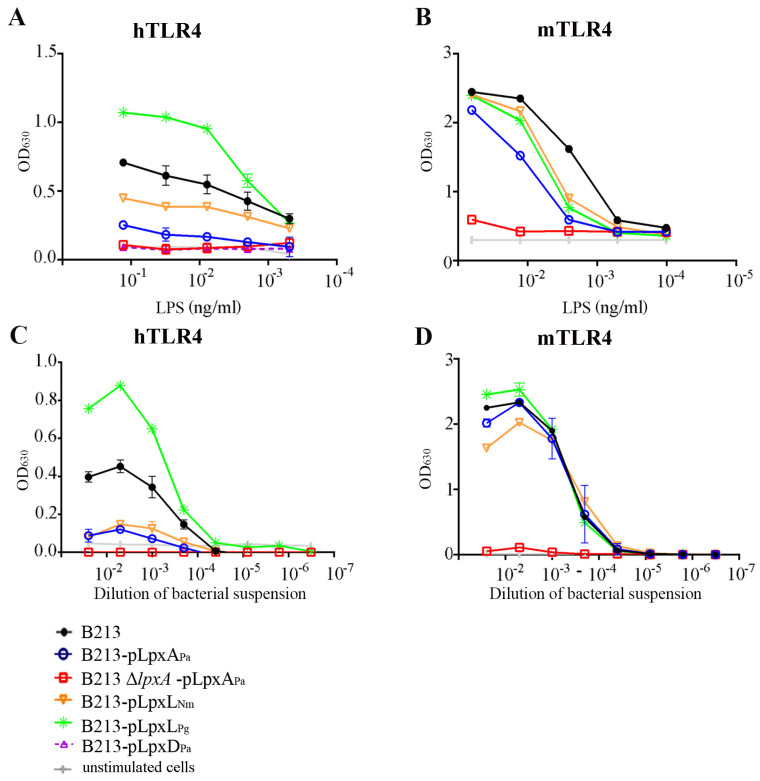
Stimulation of HEK293-Blue cells expressing hTLR4 (**A**,**C**) or mTLR4 (**B**,**D**) with purified LPS (**A**,**B**) or whole-cell preparations of B213 and derivatives (**C**,**D**). LPS preparations and bacterial suspensions were serially diluted. After incubation for 2 h with HEK293-Blue cells expressing mTLR4 or for 4 h with HEK293-Blue cells expressing hTLR4, alkaline phosphatase activity was determined. Graphs show the mean and standard deviation from a representative experiment of three repeats in duplicate.

**Figure 4 vaccines-08-00594-f004:**
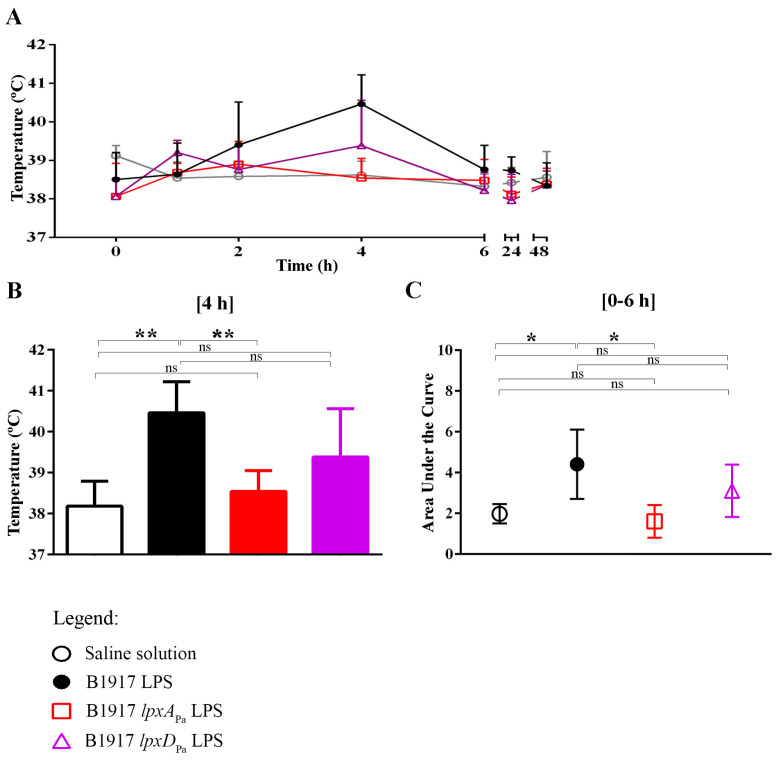
Pyrogenicity assays in rabbits. Groups of five animals were injected with saline solution (control) or with 10 µg of *B. pertussis* B1917 wild-type LPS, or mutant LPS derivatives resulting from the expression of *lpxA*_Pa_ or *lpxD*_Pa_. The temperature of each animal was monitored at different time intervals after injection. (**A**) Mean body temperature and standard deviation for each group before injection (0 h) and at different times post-injection are shown (1, 2, 4, 6, 24, 48 h). (**B**) The differences in body temperatures within groups at 4 h post-injection. Statistically significant differences were determined using ANOVA and Dunnett tests and are indicated with two asterisks (*p* < 0.001). ns, not significant. (**C**) The mean of area under the curve and standard deviation of all animals of a group were calculated between 0 and 6 h using as baseline the mean of each group. Statistically significant differences between the two groups are indicated with one asterisk (*p* < 0.05) using an unpaired *t*-test. Ns, not significant.

**Figure 5 vaccines-08-00594-f005:**
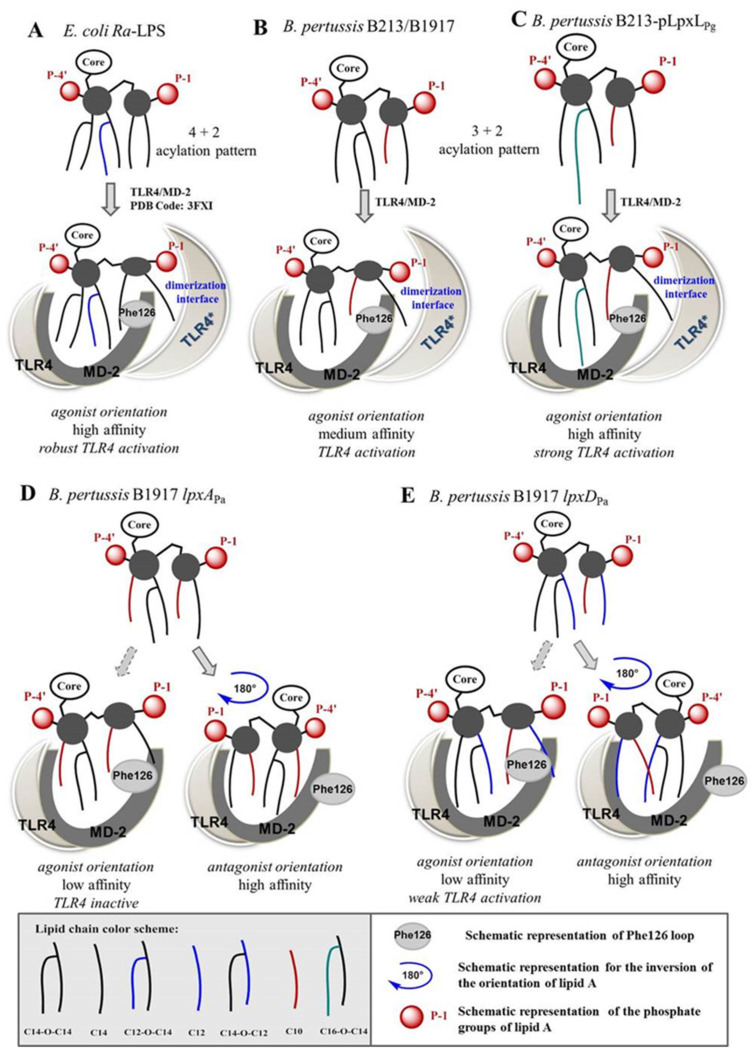
The proposed mode of interaction of LPS with hTLR4/MD-2. The proposed binding modes of LPS of *E. coli* K-12 (PDB code: 3FXI) (**A**), *B. pertussis* B213/B1917 (**B**), and several mutant LPS derivatives generated in this study (**C**–**E**) are shown. The binding poses (orientation and affinity) with MD-2 are displayed below the figures.

**Table 1 vaccines-08-00594-t001:** Plasmids and strains used in this study.

Plasmids/Strains	Characteristics ^a^	References
**Plasmids**		
pMMB67EH	Broad host-range vector, P*tac*, *lacI^q^*, Amp^R^	[30]
pKAS32	Allele-exchange suicide vector, Amp^R^	[31]
pMMB67EH-PagL_Pa_	pMMB67EH harboring *pagL* from *P. aeruginosa* PAO1	[8]
pMMB67EH-LpxA_Pa_	pMMB67EH harboring *lpxA* from *P. aeruginosa* PAO1	This study
pMMB67EH-LpxL_Nm_	pMMB67EH harboring *lpxL1* from *N. meningitidis* H44/76	This study
pMMB67EH-LpxL_Pg_	pMMB67EH harboring *lpxL* from *P. gingivalis* ATCC33277	This study
pMMB67EH-LpxD_Pa_	pMMB67EH harboring *lpxD* from *P. aeruginosa* PAO1	This study
pKA32-EFGH-*lpxL_2_*::*gm*	pKAS32 derivative harboring DNA segments of *lpxL_1_* and downstream of *lpxL_2_* separated by a *gm* cassette, Amp^R^, Gm^R^	Geurtsen J
pKO*lpxL_2_*::*gm* (a)	pKA32-EFGH-*lpxL_2_*::*gm* derivative harboring a DNA segment upstream of *lpxL_1_* and the entire *lpxL_2_* gene except for the first 6 nucleotides together with a partial *dap* gene with, in between a *gm* cassette in the same orientation as the operon*,* Amp^R^, Gm^R^	This study
pKO*lpxL_2_*::*gm* (b)	Like pKO*lpxL_2_*::*gm* (a) but with the *gm* cassette in the opposite orientation, Amp^R^, Gm^R^	This study
pRTP113368K2a	pKAS32 derivative harboring DNA segments of *lpxL_2_*, Amp^R^, Kan^R^ (*kan* in similar orientation as the operon)	Hamstra HJ
pRTP113368K1a	pRTP113368K2a derivative harboring *kan* in opposite orientation as the operon.	Hamstra HJ
pRT669	*lpxA* knockout construct, Amp^R^, Kan^R^ (*kan* in opposite orientation as the *lpxA* gene)	Hamstra HJ
pSS1129	Suicide vector for allelic exchange, Gm^R^, Amp^R^	[32]
pC*lpxA*_Pa_	pSS1129 derivative for replacement of *lpxA* gene of *B. pertussis* by *lpxA* gene of *P. aeruginosa*	This study
pC*lpxD*_Pa_	Construct for replacement of *lpxD* gene of *B. pertussis* by *lpxA* gene of *P. aeruginosa*	This study
**Strains**		
*Escherichia coli*		
DH5α	F^−^, Δ(*lacZYA-argF*)*U169 thi-1 hsdR17 gyrA96 recA 1 endA 1 supE44 relA1 phoA Φ80 dlacZΔM15*	UU lab collection
SM10λpir	*thi thr leu fhuA lacY supE recA*::RP4–2-Tc::Mu *λpir* R6K Kan^R^	[33]
BL21(DE3)	Contains gene for T7 DNA polymerase	Invitrogen
BL21-pLpxA_Pa_	BL21(DE3) carrying pMMB67EH-LpxA_Pa_	This study
BL21-pLpxL_Nm_	BL21(DE3) carrying pMMB67EH-LpxL_Nm_	This study
BL21-pLpxL_Pg_	BL21(DE3) carrying pMMB67EH-LpxL_Pg_	This study
BL21-pLpxD_Pa_	BL21(DE3) carrying pMMB67EH-LpxD_Pa_	This study
*Bordetella pertussis*		
B213	Str^R^ derivative of strain Tohama I, Nal^R^	[27]
B213-pLpxA_Pa_	B213 carrying pMMB67EH-LpxA_Pa_	This study
B213 Δ*lpxA*-pLpxA_Pa_	B213 with inactivated *lpxA* gene carrying pMMB67EH-LpxA_Pa_	This study
B213-pLpxL_Nm_	B213 carrying pMMB67EH-LpxL_Nm_	This study
B213-pLpxL_Pg_	B213 carrying pMMB67EH-LpxL_Pg_	This study
B213-pLpxD_Pa_	B213 carrying pMMB67EH-LpxD_Pa_	This study
B1917		[28]
B1917 *lpxA*_Pa_	B1917 with *lpxA* gene replaced by *lpxA* from *P. aeruginosa*	This study
B1917 *lpxD*_Pa_	B1917 with *lpxD* gene replaced by *lpxD* from *P. aeruginosa*	This study

^a^ P*tac*, *tac* promoter; Amp^R^, ampicillin-resistant; Gm^R^, gentamicin resistant, Kan^R^, kanamycin-resistant; Nal^R^, Nalidixic acid-resistant; Str^R^, streptomycin-resistant.

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
