# Peer review of "Shortening the Lipid A Acyl Chains of *Bordetella pertussis* Enables Depletion of Lipopolysaccharide Endotoxic Activity"

_vaccines, 2020, doi:10.3390/vaccines8040594_

Round 1
Reviewer 1 Report
This paper is about re-engineering Lipid A to reduce toxicity in new vaccine formulations of B pertussis, with the goal of generating new cellular vaccines. The authors focused on the acyl chain number and length variability between different species to explore the possible configurations. An important observation from this work is that activation of the hTLR4 correlates with the length of the secondary acyl chain in the order C16 > C14 > C12.
This paper is very well written and informative. Figures and diagrams are excellent. However, the following concerns and suggestions should be considered before publication:
Interestingly, authors were not able to inactivate the Lpx l2 gene, which speaks about the critical need for this gene for the lab growth of B pertussis. Since the C16 LPS substitution did not affect growth but significantly changed the NFKB response, is it possible the availability of the C16 FFA to the bacteria is essential for their survival and expansion? Please clarify.
The pyrogenicity response experiments were not performed in mice. How similar is the rabbit receptor complex to human and mouse TLR4?
Since the stimulation of TLR4 was particularly good for the LPS mutants with C16 chains, what could have been the results of the pyrogenicity tests with these mutated LPS molecules? Clarification is needed.
Please provide more detail about the statistical tests used in the methods section.
Author Response
This paper is about re-engineering Lipid A to reduce toxicity in new vaccine formulations of B pertussis, with the goal of generating new cellular vaccines. The authors focused on the acyl chain number and length variability between different species to explore the possible configurations. An important observation from this work is that activation of the hTLR4 correlates with the length of the secondary acyl chain in the order C16 > C14 > C12.
This paper is very well written and informative. Figures and diagrams are excellent. However, the following concerns and suggestions should be considered before publication:
Interestingly, authors were not able to inactivate the Lpx l2 gene, which speaks about the critical need for this gene for the lab growth of B pertussis. Since the C16 LPS substitution did not affect growth but significantly changed the NFKB response, is it possible the availability of the C16 FFA to the bacteria is essential for their survival and expansion? Please clarify.
Response: I’m sorry, but we don’t really understand the question. Probably, there is some misunderstanding. The bacteria are not dependent on the availability of any fatty acids in the growth medium, but they synthesize fatty acids of all chain lengths needed themselves. The acyl transferases, such as LpxL2, select fatty acids with a specific chain length from the fatty-acid elongation pathway and incorporate them into lipid A. Wild-type B. pertussis does not incorporate C16 chains in lipid A, so the presence of C16 FFA in lipid A is not essential for bacterial survival. C16 FFA are incorporated in the phospholipids of wild-type B. pertussis, and it is very well possible (but very difficult to address experimentally) that the bacteria cannot survive without C16 chains in their phospholipids. However, LpxL2 is not implicated in the phospholipid synthesis pathway.
The pyrogenicity response experiments were not performed in mice. How similar is the rabbit receptor complex to human and mouse TLR4?
Response: As stated in the text (lines 396-398), we have selected the rabbit model as it is accepted as standard model for determining pyrogenicity in most pharmacopoeias. Indeed, human TLR4 is more similar to rabbit TLR4 than to mouse TLR4 as we have added to the text (lines398- 401)
Since the stimulation of TLR4 was particularly good for the LPS mutants with C16 chains, what could have been the results of the pyrogenicity tests with these mutated LPS molecules? Clarification is needed.
Response: Obviously, one would expect the pyrogenicity of the mutant LPS with C16 chains to be increased. Animal experiments, however, are highly regulated by an ethical committee. Our proposal was to reduce the activity of pertussis LPS, and we received permission to test candidates in animal experiments. Testing variants with higher activity serves no purpose in vaccine development and is, therefore, not acceptable to the committee.
Please provide more detail about the statistical tests used in the methods section
Response: Information of the statistical analyses applied has been expanded and is presented now in a separate section (lines 232-253).
Reviewer 2 Report
The authors sought to reduce the endotoxicity of whole-cell B. pertussis vaccines by lipid A engineering. I felt that a lot of effort was contributed to achieve this goal, and this manuscript is clear logic and well-written. However, it is mandatory to explain/correct the manuscript in some points.
- Several important papers should be included in the introduction to provide sufficient background on the production of cytokines through TLR (line 54)- for example, Journal of Neuroinflammation 15, 202 (2018); Journal of Proteome Research 19 (6), 2236-2246 (2020).
- Figure 2. Is it positive or negative ESI?
- What is the control in the RT-PCR experiment?
Author Response
The authors sought to reduce the endotoxicity of whole-cell B. pertussis vaccines by lipid A engineering. I felt that a lot of effort was contributed to achieve this goal, and this manuscript is clear logic and well-written. However, it is mandatory to explain/correct the manuscript in some points.
Several important papers should be included in the introduction to provide sufficient background on the production of cytokines through TLR (line 54)- for example, Journal of Neuroinflammation 15, 202 (2018); Journal of Proteome Research 19 (6), 2236-2246 (2020).
Response: In line 54, we mention the possible reasons of Bordetella vaccine failure. Background information on the production of cytokines through TLR is pointed out in lines 52-55 with several references. The references mentioned by the referee both deal with effects of docosahexaenoic acid on LPS-stimulated microglial cells which, although interesting, is not particularly relevant to our studies.
Figure 2. Is it positive or negative ESI?
Response: It was negative ion ESI-MS as stated in Materials and Methods (line 193) and in the legend to Figure 2.
What is the control in the RT-PCR experiment?
Response: The negative control in the RT-PCR experiments shown in Fig. S1A is a strain with plasmid pMMB67EH-PagLPa. This plasmid contains the same vector as the other plasmids, but with an irrelevant insert (the pagL gene of P. aeruginosa). Hence, the ampC gene gives a positive signal and the lpxA and lpxL genes are negative. The plasmid is described in Table 1, and the legend to Fig. S1A is extended for clarification.
Reviewer 3 Report
Overall the paper is a well written account of the modification of the Bordatella pertussis Lipid A portion of LPS with altered acyl chain lengths to effect different physiological outcomes in cell and rabbit models. The paper illustrates the production of LPS molecules with altered acyl chains on the glucosamines of Lipid A using lpxL, lpxA and LpxD variants from other bacteria. The authors then demonstrate in human cells and rabbits a reduction in endotoxicity, with the B1917 lpxAPa LPS causing similar effect on rabbit termperature as compared to saline injection. They also demonstrated that while some variants decreased TLR4 activation, the lpxLPg variant increased TLR4 activation. The results suggest that B. pertussis cell variants with altered acyl chains could be used for pertussis vaccines that have reduced reactogenicity as compared to their WT progenators, while maintaining activity.
Overall, the paper was compelling, however the methods need expansion (as indicated below), and the corresponding text of the results sections could benefit from greater explanations for the reader to aid understanding.
Lines 126-140 - your description of methods for plasmid generation is not immediately clear, and would benefit from added details.
Lines 175-177 - you indicate that you grew the B213-pLpxLNm strain twice in your series, so likely a different strain was intended, and perhaps this is a cut-paste error; did you mean B213-pLpxLPg?
Lines 180-187 - your brief protocol glosses over the cleavage previously reported by the authors in paper 26, which explains why only the Lipid A portion of the molecule is detected by MS. It is important to reiterate that the ionization method causes the "rupture of the labile linkage between the nonreducing lipid A glucosamine and 3-deoxy-d-manno-oct-2-ulosonic acid". This important detail will help readers with knowledge of LPS biosynthesis, but not the details of this MS technique.
Author Response
Overall the paper is a well written account of the modification of the Bordatella pertussis Lipid A portion of LPS with altered acyl chain lengths to effect different physiological outcomes in cell and rabbit models. The paper illustrates the production of LPS molecules with altered acyl chains on the glucosamines of Lipid A using lpxL, lpxA and LpxD variants from other bacteria. The authors then demonstrate in human cells and rabbits a reduction in endotoxicity, with the B1917 lpxAPa LPS causing similar effect on rabbit termperature as compared to saline injection. They also demonstrated that while some variants decreased TLR4 activation, the lpxLPg variant increased TLR4 activation. The results suggest that B. pertussis cell variants with altered acyl chains could be used for pertussis vaccines that have reduced reactogenicity as compared to their WT progenators, while maintaining activity.
Overall, the paper was compelling, however the methods need expansion (as indicated below), and the corresponding text of the results sections could benefit from greater explanations for the reader to aid understanding.
Lines 126-140 - your description of methods for plasmid generation is not immediately clear, and would benefit from added details.
Response: Please note that the description of plasmid constructions also included lines 141-165 of the original manuscript and is covered by lines 120-161 in the revised version. We have now added further details of the constructions to this section in the manuscript, and we transferred Table S1 to the main text as Table 1 to facilitate understanding plasmid generation.
Lines 175-177 - you indicate that you grew the B213-pLpxLNm strain twice in your series, so likely a different strain was intended, and perhaps this is a cut-paste error; did you mean B213-pLpxLPg?
Response: Thanks for pointing out this mistake, which we have corrected (now line 191).
Lines 180-187 - your brief protocol glosses over the cleavage previously reported by the authors in paper 26, which explains why only the Lipid A portion of the molecule is detected by MS. It is important to reiterate that the ionization method causes the "rupture of the labile linkage between the nonreducing lipid A glucosamine and 3-deoxy-d-manno-oct-2-ulosonic acid". This important detail will help readers with knowledge of LPS biosynthesis, but not the details of this MS technique.
Response: Thank you for the suggestion. We have included the requested information (lines 200-202).
Reviewer 4 Report
Arenas and colleagues attempt a series of genetic manipulations of Bordella pertussis in order to modify LPS structure. The hope is to moderate reactogenicity of whole cell based vaccines. Main findings are strains that remain vital producing lipid A with altered structures and lower activity toward TLR4 as well as pirogenicity.
The manuscript is well organized, written in fluent English and the experiment well designed. Thus, the manuscript is interesting for publication.
A few concerns remain:
- Are the strains used suitable (at least in principle) for the production of whole cell cells vaccines or the preparation of vesicles for immunization (as in Sci Rep. 2020 Apr 30;10(1):7396)? This issue should be mere extensively discussed.
- At page 12 authors concluded that “In conclusion, in contrast to the wild-type LPS, both mutant LPS forms elicited no signs or minimal signs of toxicity in rabbits, which is in agreement with the in vitro TLR4 stimulation assays.” The authors only reported temperature measures, this is a pirogenicity assay, not a toxicity assay!
- The authors, basing on literature, propose a structural explanation for the reduced activity of the lipidA structures found on TLR4. This model is very interesting but highly speculative as is not supported by docking simulations or experimental evidences.
Author Response
Arenas and colleagues attempt a series of genetic manipulations of Bordella pertussis in order to modify LPS structure. The hope is to moderate reactogenicity of whole cell based vaccines. Main findings are strains that remain vital producing lipid A with altered structures and lower activity toward TLR4 as well as pirogenicity.
The manuscript is well organized, written in fluent English and the experiment well designed. Thus, the manuscript is interesting for publication.
A few concerns remain:
Are the strains used suitable (at least in principle) for the production of whole cell cells vaccines or the preparation of vesicles for immunization (as in Sci Rep. 2020 Apr 30;10(1):7396)? This issue should be mere extensively discussed.
Response: Thank you for the suggestion. The strains could be used for the generation of both whole-cell vaccines or OMV-based vaccines as was already mentioned in the Conclusions section. We expanded on this issue in the discussion in lines 533-538.
At page 12 authors concluded that “In conclusion, in contrast to the wild-type LPS, both mutant LPS forms elicited no signs or minimal signs of toxicity in rabbits, which is in agreement with the in vitro TLR4 stimulation assays.” The authors only reported temperature measures, this is a pirogenicity assay, not a toxicity assay!
Response: Right and we have adapted the text accordingly (line 419). However note that one of the main signs of a LPS toxicity is fever.
The authors, basing on literature, propose a structural explanation for the reduced activity of the lipidA structures found on TLR4. This model is very interesting but highly speculative as is not supported by docking simulations or experimental evidences
Response:
TLR4/MD2 complex is an extremely flexible system, especially the co-receptor protein MD-2 undergoes substantial rearrangements upon binding of lipid A/LPS, and these rearrangements alone determine whether the system dimerizes or not. Docking studies use protein geometries obtained from crystal structures and mostly apply “flexible ligand” settings (only the ligand is considered flexible, the receptor remains rigid) or “induced fit” docking (moveable side chains in the binding pocket) which in no way describes the dynamics of TLR4/MD-2/lipid A complex, where both the protein and the ligand possess multiple degrees of flexibility. “Full flexibility” model is rarely applied in docking studies as computationally inaccurate and very expensive. Although MDS of a large protein-ligand system is tremendously computational power- and memory-consuming, it is the only computation approach which can provide decent prediction of TLR4-dependent lipid A activity. The MDS of TLR4/MD-2 bound Bp lipid A (we have to analyse four Bp LPS structures, each in two possible orientation) is a huge project which requires immense computational (and man-) power. We have to consider computational costs and runtime (3-4 months for 20 ns! x 8 systems = 2.5 years of calculation time only). Nevertheless, MDS would be again an approximation with 50% reliability (from the biological point of view) since it cannot reproduce the action of CD14, which “delivers” LPS to the binding pocket of MD-2 and is supposed to be responsible for a proper positioning of lipid A (+/- 180°) in the binding pocket of MD-2. Instead, the ligand must be “manually” placed into the binding pocket. We are still running some preliminary simulations, but this project is out of scope of the current manuscript.
Round 2
Reviewer 2 Report
All concerns were addressed.